# Outdoor Thermal Comfort of Urban Park—A Case Study

**Lili Zhang** [1,2,*], **Dong Wei** [1], **Yuyao Hou** [1], **Junfei Du** [1], **Zu'an Liu** [1], **Guomin Zhang** [2] **and Long Shi** [2]

[1] College of Architecture and Urban-Rural Planning, Sichuan Agricultural University, Dujiangyan, Sichuan 611830, China; s20172805@stu.sicau.edu.cn (D.W.); houyuyao@stu.sicau.edu.cn (Y.H.); dujunfei@stu.sicau.edu.cn (J.D.); liuzuan@stu.sicau.edu.cn (Z.L.)

[2] Civil and Infrastructure Engineering Discipline, School of Engineering, RMIT University, Melbourne, VIC 3000, Australia; kevin.zhang@rmit.edu.au (G.Z.); shilong@mail.ustc.edu.cn (L.S.)

[*] Correspondence: 41414@sicau.edu.cn

**Abstract:** Urban parks are an important component of urban public green space and a public place where a large number of urban residents choose to conduct outdoor activities. An important factor attracting people to visit and stay in urban parks is its outdoor thermal comfort, which is also an important criterion for evaluating the liability of the urban environment. In this study, through field meteorological monitoring and a questionnaire survey, outdoor thermal comfort of different types of landscape space in urban parks in Chengdu, China was studied in winter and summer. Result indicated that (1) different types of landscape spaces have different thermal comforts, (2) air temperature is the most important factor affecting outdoor thermal comfort; (3) because the thermal sensation judgment of outdoor thermal comfort research in Chengdu area, an ASHRAE seven-sites scale can be used; (4) the neutral temperature ranges of Physiological Equivalent Temperature (PET) and Universal Thermal Climate Index (UTCI) in Chengdu in winter and summer were obtained through research; (5) and UTCI is the best index for evaluating outdoor thermal comfort in Chengdu. These findings provide theoretical benchmarks and technical references for urban planners and landscape designers to optimize outdoor thermal comfort in urban areas to establish a more comfortable and healthy living environment for urban residents.

**Keywords:** outdoor thermal comfort; urban park; landscape spaces; thermal index

## 1. Introduction

Under the background of global warming and the acceleration of China's urbanization process, the climate of the city has undergone tremendous changes. The increase in urbanization rate is accompanied by the expansion of urban construction land, which has led to a sharp decline in the urban public green area, and the urban heat island effect has been continuously enhanced. The urban heat island effect [1,2] is a special local temperature distribution phenomenon that occurs simultaneously with the development of the city. Its performance characteristics are that the urban temperature is significantly higher than that of the suburbs. The increase in temperatures will have serious impact on the health of urban residents. Studies have shown that high temperatures can lead to fatigue, dizziness, increased breathing, and increased heart rate [3]. More serious situations can even endanger life and cause death [4]. An urban park is a main place for outdoor activities of citizens and also an important part of urban green space system, which plays an important role in mitigating the urban heat island effect and improving the outdoor thermal comfort. Urban parks are also conducive to

energy conservation because, if a large number of urban residents can stay in the park, it reduces the use of air conditioning equipment [5].

Outdoor thermal comfort is the most important factor to attract urban residents to urban parks. Thermal comfort is defined as the "condition of mind that expresses satisfaction with the thermal environment and is assessed by subjective evaluation" [6]. In recent years, with the deterioration of urban climate, people pay more and more attention to the urban environment, which makes researchers all over the world focus on outdoor thermal comfort, and the number of research results about outdoor thermal comfort is increasing year by year. The outdoor environment has many factors affecting thermal comfort, such as (1) meteorological factors—air temperature, relative humidity, wind speed and direction, solar radiation, etc.—and (2) personal factors—gender, age, length of stay, length of residence, etc. These factors are the key to studying outdoor thermal comfort. Many domestic and foreign scholars have carried out a lot of research on these factors. The previous research literature indicates that the method of studying outdoor thermal comfort is mainly simulation analysis [7–9] and on-site measurement [10–12]. Nikolopoulou et al. [13,14] pointed out in published papers that field monitoring is the main method for evaluating outdoor thermal comfort. Field monitoring is the use of relevant measuring instruments for the study site for outdoor meteorological parameters (such as air temperature—$T_a$, relative humidity—$RH$, black globe temperature—$T_g$, wind speed—$v$, global solar radiation—$G$, etc.), and subjective questionnaire surveys are conducted near the monitoring sites [15–18]. The field monitoring outdoor thermal comfort evaluation method is to find out the comfortable or acceptable thermal conditions for local residents, to understand human body perception of the thermal environment [19–21], and to provide valuable reference for urban planners and park designers [9,22–25], so as to formulate strategies for optimizing the outdoor thermal environment.

China is a large country, with a land area of about 9.6 million square kilometers. Because of its vast area, its climate has great diversity and complexity, and this has led to the formation of many regions with distinctive climate characteristics. Different climatic conditions place different requirements on building construction. In order to meet the needs of ventilation, shading, heat insulation in hot areas, heating, anti-freezing and heat insulation in cold areas, and to clarify the scientific connection between the building and the climate, China has formulated the "Code for Thermal Design of Civil Buildings". These specifications are based on the average temperature of the coldest and hottest months of the year and divide the whole of China into five climatic regions—the severe cold zone, cold zone, hot-summer and warm-winter zone, hot-summer and cold-winter zone, and temperate zone [26,27] (see Figure 1.). Among them, the location of our study object is located in the hot-summer and cold-winter zone. Its climatic characteristics are that the average temperature of the coldest month meets 0–10 °C, and the average temperature of the hottest month meets 25–30 °C [27].

Table 1 lists various outdoor thermal comfort studies conducted in China in the past decade. At present, most of the research on outdoor thermal comfort is concentrated in the hot-summer and warm-winter climate zone, and the representative cities are Guangzhou [10,28,29] and Hong Kong [12,30,31]. There are the second largest number of studies from the severe cold zone and cold zone, and the representative cities are Harbin [32] and Xi'an [33,34] respectively. There is less concern for summer-hot and winter-cold climate zone, such as Chengdu. Therefore, previous research results may not be applicable to the hot-summer and cold-winter zone.

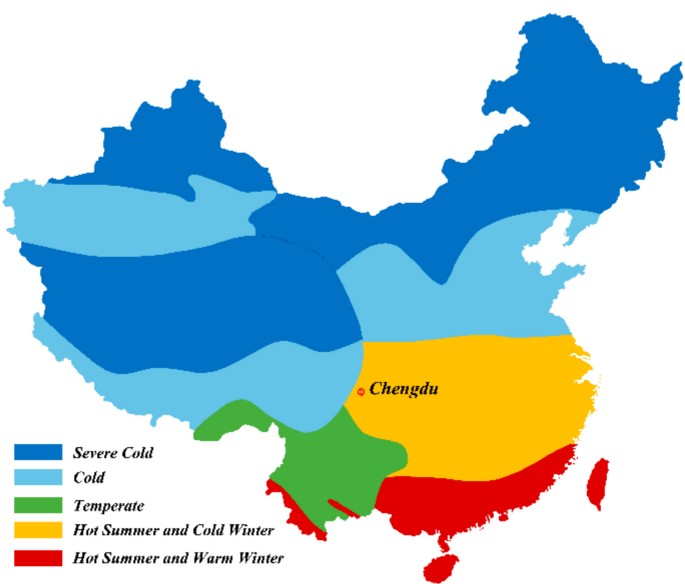

**Figure 1.** The climate classification of Chengdu.

In recent studies, Physiologically Equivalent Temperature (PET) [11,12,28] and Universal Thermal Climate Index (UTCI) are the most commonly used indicators of outdoor thermal comfort (PET: 20 out of 28 in Table 1, UTCI: 7 out of 28 in Table 1) because both indicators integrate meteorological parameters (air temperature, relative humidity, wind speed, and average radiant temperature) and personal factors (clothing and metabolic rate). The results are calculated by the relevant formulas, so the two indicators are mainly used in unstable outdoor environments. And the units of both indicators are in °C, which is convenient and intuitive for comparison and analysis. A large number of related studies using PET and UTCI as indicators for outdoor thermal comfort [28,29,35–40]. Min et al. [33] evaluate the thermal comfort of outdoor human body in Xi'an. A study conducted by Pui Kwan Cheung and C.Y. Jim [35] in Hong Kong showed that PET was more sensitive to human thermal response changes than UTCI in the hot summer. These studies show that the applicability of various indicators is different under different climatic conditions. Therefore, each different climate area needs to carry out relevant research in order to clarify the characteristics of outdoor thermal comfort in this area.

There are many kinds of outdoor and comfortable research objects, such as urban blocks [11,41], university campuses [12,37], urban residential communities [28,42], and urban parks [33–35]. Urban parks are currently the focus of outdoor thermal comfort research because of the slowdown in urban heat island effects and the improvement of urban thermal environments. Existing related research has determined the impact of individual elements of urban parks on the outdoor thermal environment and human thermal comfort. Xu et al. [43] found that people's thermal comfort in the plant environment is better than in water and built environments. Klemn et al. [44] found that grass and water bodies mitigate the effects of thermal discomfort, while thermal comfort is affected by the shape and height of the canopy. Mahmoud [45] found that the sky view factor (SVF) and wind speed (*v*) have an enormous impact on outdoor human thermal comfort. The study concluded that different landscape elements have different effects on human thermal comfort. These studies have shown that the human body's thermal sensation and thermal comfort are affected by landscape elements such as vegetation, water bodies, and buildings. However, most of the research is focused on a single landscape element without considering the impact of the landscape and its combination on human thermal comfort.

**Table 1.** Summary of outdoor thermal comfort research in China in the past decade.

| Authors | City | Climate [1] | Type | Year [2] | Season | Index | Ref. |
|---|---|---|---|---|---|---|---|
| T. Lin, et al. | Taiwan | HSWW | University campus | 2004–2005 | Winter, Summer | SET*[3] | [46] |
| E. Ng, V. Cheng | Hong Kong | HSWW | Urban block | 2006–2007 | Summer, Winter | PET | [47] |
| T. Xi, et al. | Guangzhou | HSWW | University campus | 2010 | July | SET *, MRT [3] | [10] |
| D. Lai, et al. | Wuhan | HSCW | Urban residential community | 2011 | August to November | TSV [3] | [5] |
| X. He, et al. | Beijing | C | Urban block | 2011 | December | PET | [48] |
| J. Huang, et al. | Wuhan | HWCW | Urban residential community | 2011–2014 | All year | UTCI | [49] |
| D. Lai, et al. | Tianjin | C | Urban park | 2012 | All year | UTCI, PET | [11] |
| Y. Zeng, L. Dong | Chengdu | HSCW | Urban block | 2012 | August | $T_{mrt}$, PET | [50] |
| W. Liu, et al. | Changsha | HSCW | University campus | 2012–2013 | March 2012 to December 2013 | PET | [51] |
| B. Cao, et al. | Shenzhen | HSWW | Urban block | 2013 | September | TSV | [52] |
| J. Yao, et al. | Shanghai | HSCW | Urban block | 2013–2014 | December 2013 to February 2014 | TSV | [53] |
| L. Zhao, et al. | Guangzhou | HSWW | University campus | 2014 | August to mid-October | SET* | [29] |
| J. Niu, et al. | Hong Kong | HSWW | University campus | 2014 | June | PET | [12] |
| L. Chen, et al. | Shanghai | HSCW | Urban park | 2014–2015 | November 2014 to January 2015 | PET | [41] |
| K. Li, et al. | Guangzhou | HSWW | Urban residential community | 2015 | Winter, Spring, Summer | PET | [28] |
| T. Huang, et al. | Hong Kong | HSWW | University campus | 2016 | Summer, Autumn, Winter | PET, UTCI | [30] |
| J. Li, et al. | Hong Kong | HSWW | University campus | 2016 | March to December | UTCI | [31] |
| Y. Du, et al. | Hong Kong | HSWW | University campus | 2016 | All year | PET | [54] |
| Z. Fang, et al. | Guangzhou | HSWW | University campus | 2016 | All year | PET, UTCI | [36,37,55] |
| X. Chen, et al. | Harbin | SC | University campus | 2016 | All year | PET | [32] |
| X. Ma, et al. | Taizhou | HSCW | Urban block | 2016 | July | PET | [42,56] |
| X. Ma, et al. | Fo Shan | HSWW | Urban block | 2016 | July | PET | [57,58] |
| Y. Wang, et al. | Guangzhou | HSWW | University campus | 2016–2017 | All year | $T_{op}$ [3], PET | [59] |
| S. Yin, et al. | Guangzhou | HSWW | Urban block | 2017 | July | PET | [60] |
| P. Cheung, C.Y. Jim | Hong Kong | HSWW | Urban park | 2017 | All year | PET, UTCI | [35,39,61] |
| C. Shang, et al. | Haikou | HSWW | Holiday beach | 2018 | Spring, Autumn, Winter | PET | [38] |
| B. Cheng, et al. | Mianyang | HSCW | Urban park | 2018 | Summer, Autumn, Winter | PET | [62] |
| M. Xu, et al. | Xi'an | C | Urban park | 2018 | January | PET, UTCI | [33,34] |

1: The climate is building climate demarcation of China. Code for thermal design of civil building of China (GB 50176-2016), it divides China into five climatic zones, namely Hot Summer and Warm Winter zone (HSWW), Hot Summer and Cold Winter zone (HSCW), Temperate zone (T), Cold zone (C) and Severe Cold zone (SC) [27]. 2: The year here is the year in which the data was measured. 3: SET* is Standard Effective Temperature; MRT is Mean Radiant Temperature; TSV is Thermal Sensation Vote. $T_{op}$ is Operating Temperature.

In this study, the field meteorological measurements and subjective questionnaires in winter and summer were conducted a typical hot summer and cold winter area in Chengdu, China. The objectives of this study were as follows:

1.  To study the effects of various landscape types of urban parks on human thermal comfort in order to evaluate the landscape types with the best thermal comfort in winter and summer.
2.  To determine the neutral temperature and neutral temperature range of outdoor thermal comfort in Chengdu and evaluate the applicability of PET and UTCI of outdoor environment in Chengdu.
3.  To provide a reference level or scope for urban planners and landscape designers.

## 2. Methodology

### 2.1. Urban Park Status and Test Sites Layout

Chengdu is located in the southwest of China and the central part of Sichuan Province (see Figure 2). The Köppen climate classification of Chengdu is subtropical monsoon humid climate (Cwa) [50], and it is a hot summer and cold winter zone in according to the Code for Thermal Design of Civil Building [16]. It has the characteristics of an early spring, hot summer, cool autumn, and warm winter. The annual average temperature is 16 °C, the annual rainfall is about 1000 mm, and the maximum wind direction is calm. The climate of Chengdu is characterized by cloud and fog, humid air, and short sunshine time. The annual average temperature is the highest in July (25.1 °C) and the lowest in January (5.4 °C). The highest temperature reached in July was 29.4 °C, and the lowest temperature reached in January was 2.6 °C, while the annual average relative humidity was between 76% and 86% (see Figure 3). The temperature in Chengdu is not high in summer (the maximum temperature is generally not more than 35 °C), but the humidity is high, and it will feel stuffy. Moreover, the temperature in winter is not too low, and because the humidity is high and the sunshine is short, it will feel very cold. Therefore, this study is focused on human body thermal comfort in winter and summer.

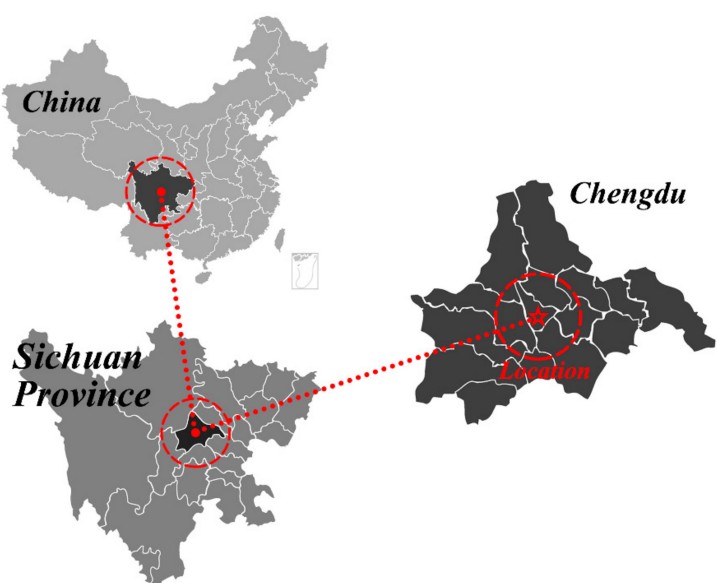

**Figure 2.** The location of Chengdu.

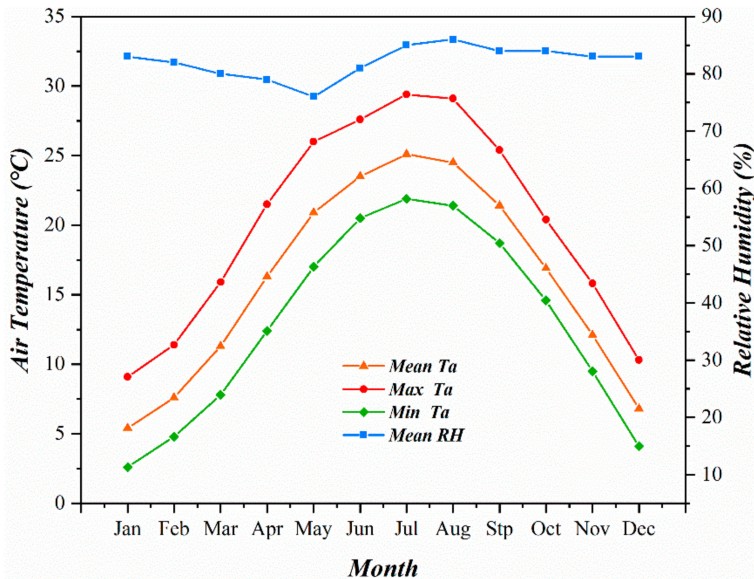

**Figure 3.** Monthly mean/maximum/minimum air temperature and mean relative humidity in Chengdu from 1981 to 2010 [63].

The park monitored in this study is located in the central part of Chengdu, which is the core part of the urban area. Chengdu People's Park is a comprehensive park integrating gardening, culture, leisure, and entertainment. It covers an area of 11.26 ha and is the largest open park in the urban center. There are many kinds of plants in the park, and the microclimate environment is diverse. There are plenty of space for public activities (square, lakeside, lawn, arbor forest, etc.). This park is an ideal place to study the outdoor thermal comfort of various landscape spaces.

Based on the field investigation and according to the difference of landscape elements combination, four different locations (Sites A–D) in the park were selected for meteorological survey and questionnaire survey (see Figure 4). Site A is located in the eastern part of the park. The landscape type is a square. The surrounding vegetation includes *Ficus microcarpa* and bamboos, and there are artificial ponds. Site B is located in the south of the park, close to the artificial landscape lake, and the nearby vegetation is bushes and willows. Site C is located in the middle of the park, and the landscape type is lawn, which is the most exposed site. Site D is located in the west of the park, where the landscape type is woods, and the vegetation coverage is extremely high. The types of trees are Nanmu, Ginkgo, and Palmetto. In order to further quantify the landscape space, fisheye images of four measurement sites in winter and summer were taken, and the Sky View Factor (SVF) of all measurement sites was calculated using RayMan software [33–35,61] (see Figure 5).

### 2.2. Meteorological Measurement

The meteorological measurements in this study were conducted from 14 to 16 January 2019 in winter from 09:00 to 18:00 and July 24–26 in summer, 07:00–20:30. The measurement date is chosen taking into account the typical coldest and hottest climate in Chengdu. The choice of measurement time is based on the most common use time of the park to ensure that the survey covers most of the people coming to the park. The measuring instruments were placed at a height of 1.5 m of the four measuring sites, and the data was recorded every 15 minutes. The measured meteorological parameters included air temperature ($T_a$), relative humidity ($RH$), black globe temperature ($T_g$), wind speed ($v$), and global solar radiation ($G$). Table 2 shows basic information such as the accuracy and range of the measuring instrument. All instruments (see Figure 6) were compliant with the ASHRAE Standard 55-2017 [6]. ASHRAE is the abbreviation of American Society of Heating, Refrigerating and Air-Conditioning Engineers.

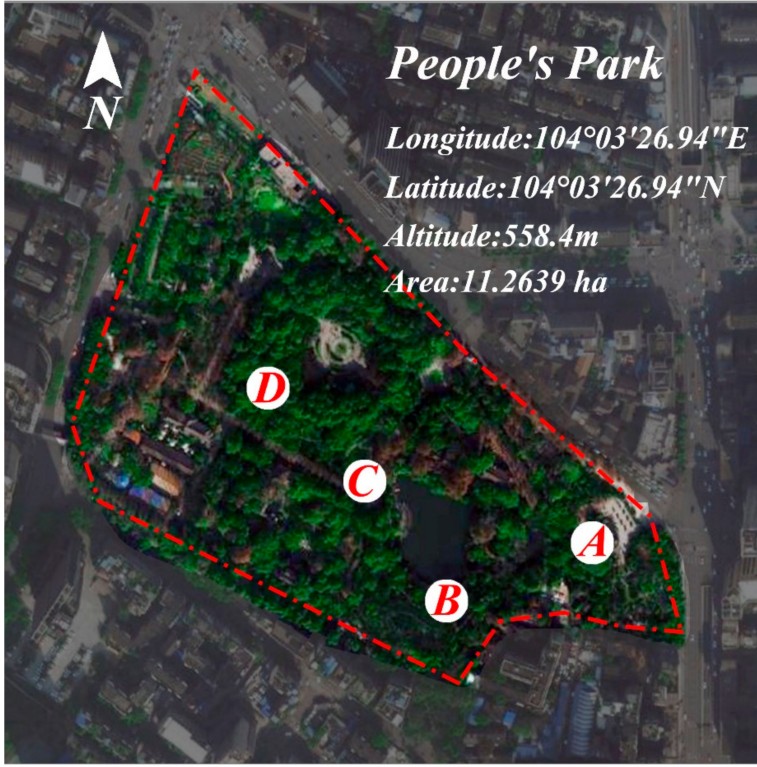

**Figure 4.** The four different measuring sites of the People's Park.

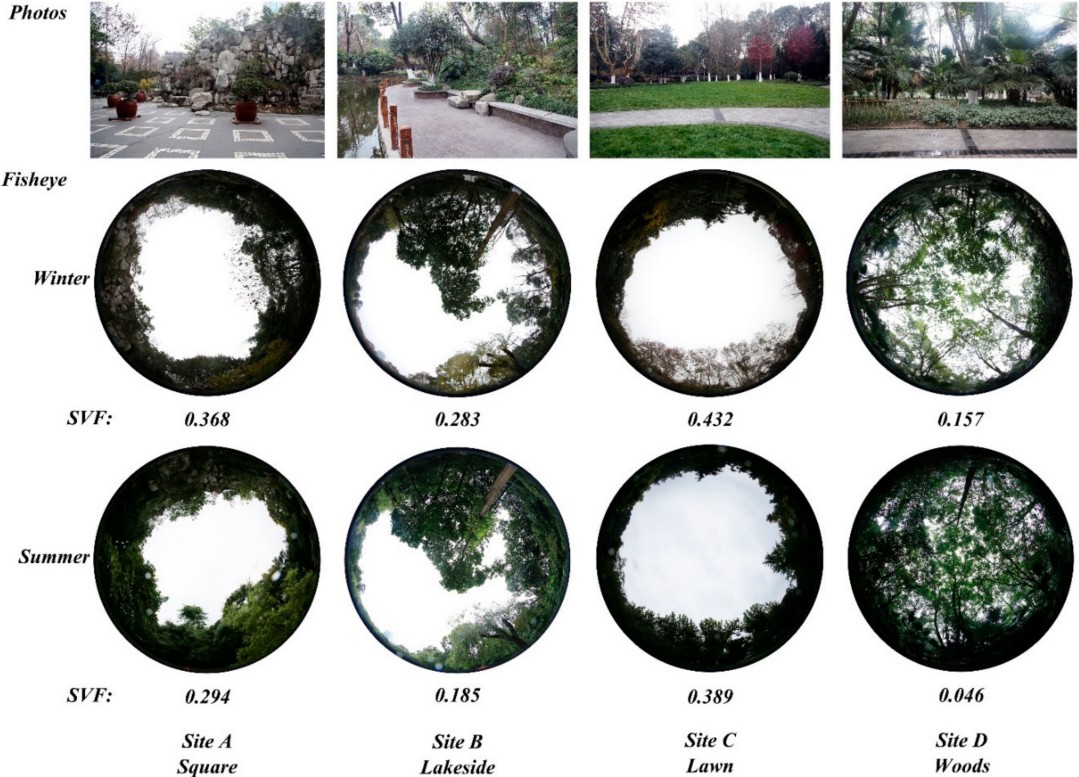

**Figure 5.** Field photos and fisheye photos of four different measuring sites.

**Table 2.** Information on experimental equipment.

| Parameter | Brand and Model | Manufacturer Location | Range | Accuracy | Resolution |
|---|---|---|---|---|---|
| Air temperature | Testo, 174H-Mini Temperature and Humidity Recorder | Schwarzwald, Germany | −20 °C to +70 °C | ±0.5 °C | 0.1 °C |
| Relative humidity | Testo, 174H-Mini Temperature and Humidity Recorder | Schwarzwald, Germany | 0 to 100 % RH | ±3 % | 0.1 % |
| Wind speed | TENMARS, TM-404 Anemometer | Taiwan, China | 0 to 25 m/s | ±2 % | 0.1 m/s |
| Black globe temperature | JT TECHNOLOGY, JTR04 Black Globe Thermometer | Beijing, China | −20 °C to 125 °C | ±0.2 °C | 0.1 °C |
| Global solar radiation | JT TECHNOLOGY, JTR05 Solar Radiometer | Beijing, China | 0 to 2000 W/m$^2$ | ≤±2 % | 1 W/m$^2$ |

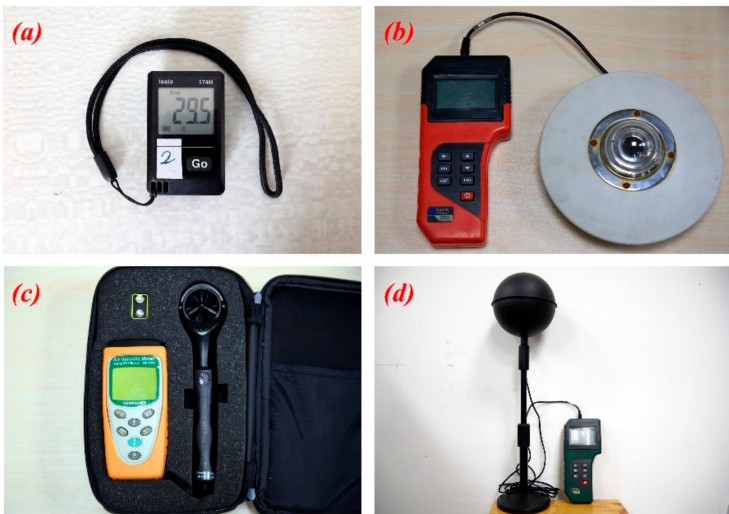

**Figure 6.** Photographs of instruments for meteorological measurements: (**a**) mini temperature and humidity recorder, (**b**) solar radiometer, (**c**) anemometer, and (**d**) black globe thermometer.

*2.3. Questionnaire Survey*

The questionnaire consists of three parts (see Figure 7). The first part is the background information of the interview, including date, time, place and activity status of the interviewees. This part is filled out by the researchers after observation. The second part is the basic information of the interviewees, including gender, age, height, weight, clothing, etc. This part is filled in by the researchers after observation and interview. The third part is human transient thermal sensation vote, which includes eight questions. Question No.1 is to verify whether the questionnaire is valid. The vote results of Question No.1 and No.4 are consistent and will be regarded as valid questionnaires. Question No. 2 is to judge thermal sensation and votes on the ASHRAE seven-point scale (i.e. −3, cold; −2, cool; −1, slightly cool; 0, neutral; 1, slightly warm; 2, warm; 3, hot). Interviewees were required to mark the value form corresponding position on the scale according to their current thermal sensation. Question No. 3.1 to No. 3.4 is the preference vote for meteorological parameters, including air temperature, relative humidity, wind speed, and solar radiation. The three-level scale is used for vote (−1, 0, and 1 respectively indicate Lower/Weak, Unchanged and Higher/Stronger). Question No. 4 is the overall thermal comfort vote, using a three-level scale (−1, uncomfortable; 0, neutral; +1, comfortable). Question No. 5 is the thermal acceptability vote with four choices (absolutely unacceptable; unacceptable; acceptable; absolutely acceptable).

The specific method of the questionnaire survey was to collect them near the four monitoring sites (an area with the monitoring equipment as the center and a radius of 5 m). The people participating in the questionnaire survey were randomly sampled. We considered the gender ratio to ensure that men and the number of women is equal. At the time of the questionnaire, we guaranteed that at least one

person participated in the questionnaire survey every 15 minutes and that the interviewees participated in each time period during the entire monitoring process. Interviewees were asked how long they had been in the monitoring sites of area. Only when the interviewees stayed at the monitoring sites for at least 20 minutes could they fill out the thermal sensation questionnaire.

A total of 419 valid questionnaires were collected, including 199 in winter and 220 in summer. Based on the field meteorological parameters measurement and questionnaire survey, the data obtained were collated and analyzed to obtain the related results of outdoor thermal comfort.

**Figure 7.** Outdoor thermal comfort questionnaire used in this study.

*2.4. Thermal Comfort Indices*

RayMan software is a tool of applied climatology developed by Dr. Andreas Matsarakis and Dr. Frank Rutz of the University of Albert Ludwig Freiburg in Freiburg, Germany. The aim of RayMan is to calculate the mean radiant temperature and different thermal indices for the quantification of thermal conditions (thermal comfort, cold stress and heat stress). By inputting air temperature ($T_a$), relative humidity (*RH*), wind speed (*v*), mean radiation temperature ($T_{mrt}$), and other meteorological parameters, the corresponding thermal indexes (such as PET, UTCI, PMV, etc.) can be calculated in order to directly evaluate the thermal comfort of an environment [64,65].

This study used PET and UTCI to quantify human thermal stress to assess outdoor thermal comfort. To date, PET and UTCI are the most widely used outdoor heat indices. PET is an index developed by Hoppe in the 1990s using the Munich Personal Energy Balance Model (MEMI), defined as an indoor or outdoor environment where human skin temperature and body temperature reach a typical indoor environment ($T_{mrt} = T_a$; $vp$ = 12hPa; $v$ = 0.1 m/s), the temperature corresponding to the equivalent thermal state. UTCI is an index developed by the International Society for Biometeorological Society (ISB) using the equivalent temperature concept, defined as a standard indoor environment that achieves the same level of comfort (*RH* = 50%; $T_{mrt} = T_a$; $v$ < 0.15m/s; $vp$ < 20hPa) and the corresponding temperature. Both PET and UTCI can integrate all meteorological parameters affecting thermal comfort or stress into a single value. The final PET and UTCI values correspond to the thermal sensation or stress levels on the PET 9 scale (see Table 3) and the UTCI 10 scale (see Table 4). Both PET and UTCI are calculated by the free software program RayMan 2.1 (http://www.urbanclimate.net/matzarakis/models/raymanpro). Both indices require input of four weather parameters ($T_a$, *RH*, *v* and $T_{mrt}$) and two personal parameters (clothing insulation and metabolic rate). The metabolic rates were set to the default level for both PET (80W) and UTCI (135W/m$^2$). The clothing insulation for both PET and UTCI was also set at standard 0.9 *clo*. Using the RayMan model to calculate the thermal index PET and UTCI requires the mean radiant temperature ($T_{mrt}$) as the input, and Calculation of $T_{mrt}$ [33,39,62] expressed as

$$T_{mrt} = \left[ \left( T_g + 273.15 \right)^4 + \frac{1.1310^8 V_a^{0.6}}{\varepsilon D^{0.4}} \left( T_g - T_a \right) \right]^{\frac{1}{4}} - 273.15 \tag{1}$$

where D is the spherical diameter (D = 150 mm in this study) and $\varepsilon$ is the globe emissivity ($\varepsilon$ = 0.95).

**Table 3.** Classification of thermal sensation and stress on the Physiological Equivalent Temperature (PET) scale.

| PET (°C) | Thermal Perception | Grade of Physical Stress |
|:---:|:---:|:---:|
| > 41 | Very hot | Extreme heat stress |
| 35 to 41 | Hot | Strong heat stress |
| 29 to 35 | Warm | Moderate heat stress |
| 23 to 29 | Slightly warm | Slight heat stress |
| 18 to 23 | Neutral (Comfortable) | No thermal stress |
| 13 to 18 | Slightly cool | Slight cold stress |
| 8 to 13 | Cool | Moderate cold stress |
| 4 to 8 | Cold | Strong cold stress |
| ≤ 4 | Very cold | Extreme cold stress |

Source: [13].

**Table 4.** Thermal stress classification for the Universal Thermal Climate Index (UTCI).

| UTCI (°C) | Thermal Stress Category |
|---|---|
| ≥ +46 | Extreme heat stress |
| +38 to +46 | Very strong heat stress |
| +32 to +38 | Strong heat stress |
| +26 to +32 | Moderate heat stress |
| +9 to +26 | No thermal stress |
| 0 to +9 | Slight cold stress |
| −13 to 0 | Moderate cold stress |
| −27 to −13 | Strong cold stress |
| −40 to −27 | Very strong cold stress |
| <−40 | Extreme cold stress |

Source: [66].

## 3. Results

### 3.1. Experimental Results

Table 5 summarizes the field measurements of meteorological parameters in winter and summer. Among the four measuring sites, the mean air temperature (Mean $T_a$) in winter and summer is Site B (winter 10.3 °C, summer 30.4 °C), The reason is that beside the Site B is a large artificial lake, the specific heat capacity of the water body is large, it can store heat and release slowly, so the average air temperature of Site B is the largest of the four points. The specific heat capacity of the water body is large, it can store heat and release it slowly, so the mean air temperature of Site B is the largest. The mean air temperature and maximum air temperature of Site D are the lowest among the four measuring sites in winter and summer. Site D is the most closed place, and its SVF (winter 0.157, summer 0.046) is the lowest among the four measuring sites. Site D also has the lowest solar radiation intensity (winter 36 W/m$^2$, summer 55 W/m$^2$). Because Site D lacks direct sunlight, it causes the air temperature to be low. The minimum air temperatures of the four sites are very similar (winter-near 5 °C, summer-near 25 °C), which indicates that the minimum air temperatures in the park are not affected by the landscape spatial differences. In relative humidity (**RH**), the data of the three sites (Site A, Site B, and Site C) are very similar in winter and summer. The mean, maximum, and minimum data of the relative humidity of Site D are the highest among the four measuring sites. The reason is that Site D has a large amount of vegetation coverage, the plant has the function of preserving moisture, and the shielding effect of the plant reduces the transpiration of water, making the relative humidity of Site D the highest at the four sites. The average wind speed of Site B in winter and summer (winter 0.3 m/s, summer 0.2 m/s), the maximum wind speed (winter 0.8 m/s, summer 0.9 m/s) is the largest. Site C is the most exposed landscape space, and its SVF is the highest of the four sites in winter and summer (winter 0.423, summer 0.389), and its maximum black global temperature is also the highest (winter 19.4 °C, summer 39.5 °C).The change of black global temperature is influenced by solar radiation. The stronger the solar radiation, the higher the black global temperature. In summary, Site B is the place with the highest of average temperature and wind speed at the four sites. The mean temperature of Site D is the lowest and the relative humidity is the highest.

**Table 5.** Meteorological parameters of four sites during the measurement period in winter and summer.

| Season | Site | $T_a$ (°C) | | | RH (%) | | | $v$ (m/s) | | | $T_g$ (°C) | | | G (W/m²) |
|---|---|---|---|---|---|---|---|---|---|---|---|---|---|---|
| | | Mean | Max | Min | Mean | Max | Min | Mean | Max | Min | Mean | Max | Min | Daily Average |
| Winter | A | 9.7 | 12.2 | 5.5 | 54.6 | 79.4 | 35.9 | 0.1 | 0.6 | 0.0 | 10.1 | 13.5 | 5.5 | 74 |
| | B | 10.3 | 16.2 | 5.3 | 53.2 | 77.0 | 34.2 | 0.3 | 0.8 | 0.0 | 10.5 | 16.4 | 4.8 | 81 |
| | C | 10.1 | 18.1 | 5.2 | 54.1 | 79.0 | 33.4 | 0.2 | 0.6 | 0.0 | 10.5 | 19.4 | 4.7 | 76 |
| | D | 8.5 | 9.8 | 5.2 | 59.4 | 77.4 | 42.9 | 0.1 | 0.4 | 0.0 | 8.7 | 10.1 | 5.3 | 36 |
| Summer | A | 30.0 | 33.8 | 25.8 | 66.0 | 84.4 | 48.9 | 0.1 | 0.8 | 0.0 | 31.5 | 39.1 | 25.5 | 123 |
| | B | 30.4 | 36.7 | 25.7 | 67.6 | 87.3 | 46.8 | 0.2 | 0.9 | 0.0 | 31.2 | 39.2 | 25.2 | 168 |
| | C | 29.6 | 35.7 | 25.2 | 68.4 | 87.5 | 46.5 | 0.1 | 0.4 | 0.0 | 31.0 | 39.5 | 25.6 | 187 |
| | D | 27.3 | 29.4 | 25.0 | 74.9 | 87.9 | 61.8 | 0.1 | 0.2 | 0.0 | 27.4 | 29.4 | 25.1 | 55 |

## 3.2. Questionnaire Survey Results

A total of 419 questionnaires were collected in this study, including 199 in winter and 220 in summer. The gender of the interviewees was evenly distributed (46.1% male,53.9% female); the average age of the interviewees was 40 for males and 43 for females; the average height was 169.7 cm for males and 163.8 cm for females; and the average weight was 68.1 kg for males and 60.1 kg for females (see Table 6). The average height and weight of the interviewees are typical of Chengdu. Table 7 summarizes the interview data of thermal sensation involved in the questionnaire in the winter and summer. In the thermal sensation vote (TSV), the most voted in the winter are "slight cool = −1" and "cool = −2," both of which are 34.2%. The most voted in summer was "neutral = 0" (38.2%), followed by "slight warm = 1" (34.1%). Most of the interviewees in the air temperature preference voted to be higher in the winter (76.4%) and lower in the summer (73.6%). In the relative humidity preference vote, the vast majority of interviewees in winter and summer wanted no change (winter 78.4%, summer 88.6%). Regarding the preference vote of wind speed, more than half of the interviewees wanted no unchanged (60.8%) in winter, while most of the interviewees wanted to increase wind speed (86.8%) in summer. Regarding the preference vote for solar radiation, most interviewees hoped that the solar radiation intensity in winter is stronger (81.4%) and in summer does not change (63.2%). During the study period, most of the interviewees that thermal comfort was neutral (winter 69.4% and summer 68.2%). Also, most of the interviewees voted acceptable for thermal of thermal acceptability issue. Overall, interviewees voted less for extremes (cold = –3, hot = +3), and the results showed that the thermal environment in Chengdu is not extreme.

**Table 6.** Summary of basic information of interviewee in questionnaire survey.

| Season | Gender | No. of Persons | Average Age | Average Height (m) | Average Weight (m) | Average Clothing (clo) |
|---|---|---|---|---|---|---|
| Winter | Male | 95 | 45 | 168.3 | 68.9 | 1.52 |
| | Female | 104 | 47 | 162.9 | 60.3 | 1.59 |
| Summer | Male | 98 | 35 | 171.1 | 67.3 | 0.45 |
| | Female | 122 | 38 | 164.6 | 59.8 | 0.58 |

**Table 7.** Summary of responses to the questions in the questionnaire survey.

| No. | Question | Variable | Option | Statistics and Percentage | | | |
|---|---|---|---|---|---|---|---|
| | | | | Winter | | Summer | |
| 1 | How do you feel at this moment? | TSV | Cold = −3 | 1 | 0.5% | 0 | 0 |
| | | | Cool = −2 | 68 | 34.2% | 0 | 0 |
| | | | Slight cool = −1 | 68 | 34.2% | 0 | 0 |
| | | | Neutral = 0 | 56 | 28.1% | 84 | 38.2% |
| | | | Slight warm = +1 | 6 | 3% | 75 | 34.1% |
| | | | Warm = +2 | 0 | 0 | 61 | 27.7% |
| | | | Hot = +3 | 0 | 0 | 0 | 0 |

**Table 7.** *Cont.*

| No. | Question | Variable | Option | Statistics and Percentage | | | |
|-----|----------|----------|--------|---------|---------|--------|--------|
| | | | | Winter | | Summer | |
| 2 | How would you prefer the air temperature to be? | $T_a$ preference | Higher = +1 | 152 | 76.4% | 0 | 0 |
| | | | Unchanged = 0 | 47 | 23.6% | 58 | 26.4% |
| | | | Lower = −1 | 0 | 0 | 162 | 73.6% |
| 3 | How would you prefer the relative humidity to be? | *RH* preference | Damper = +1 | 0 | 0 | 6 | 2.8% |
| | | | Unchanged = 0 | 156 | 78.4% | 195 | 88.6% |
| | | | Drier = −1 | 43 | 21.6% | 19 | 8.6% |
| 4 | How would you prefer the wind speed to do? | Wind preference | Stronger = +1 | 0 | 0 | 191 | 86.8% |
| | | | Unchanged = 0 | 121 | 60.8% | 29 | 13.2% |
| | | | Weaker = −1 | 78 | 39.2% | 0 | 0 |
| 5 | How would you prefer the solar radiation to do? | Sunshine preference | Stronger = +1 | 162 | 81.4% | 0 | 0 |
| | | | Unchanged = 0 | 36 | 18.1% | 139 | 63.2% |
| | | | Weaker = −1 | 1 | 0.5% | 81 | 36.8% |
| 6 | Please describe you overall comfort level | Thermal comfortable | Uncomfortable = −1 | 58 | 29.1% | 50 | 22.7% |
| | | | Neutral = 0 | 138 | 69.4% | 150 | 68.2% |
| | | | Comfortable = +1 | 3 | 1.5% | 20 | 9.1% |
| 7 | Your acceptable level for current thermal environment? | Thermal acceptability | Absolutely unacceptable = −2 | 3 | 1.5% | 0 | 0 |
| | | | Unacceptable = -1 | 55 | 27.6% | 63 | 28.6% |
| | | | Acceptable = +1 | 141 | 70.9% | 137 | 62.3% |
| | | | Absolutely acceptable = +2 | 0 | 0 | 20 | 9.1% |

*3.3. Effects of Different Landscape Spaces on Human Thermal Comfort*

3.3.1. Thermal Sensation Vote

In order to study the thermal response and thermal stress of different landscape spaces to urban residents, the respondents' thermal sensory vote data were collated and analyzed to assess the impact of landscape differences on human thermal comfort. The winter thermal sensation vote (TSV) showed (see Figure 8a) that the highest percentage (68%) of the respondents felt that "neutral" (TSV = 0) was at Site C, so the thermal comfort of Site C was the best in winter. The second highest percentage (64%) of respondents felt that "cool" (TSV = −2) was at Site D, so Site D had the worst thermal comfort. The results of Site A and Site B for the thermal sensation of "cool" (TSV = −2) were 42.59% and 26.67%, respectively; the results of the vote thermal sensation of "slightly cool" (TSV = -1) were 50% and 53.33%, and the vote results were very similar. The result of the thermal sensation of "neutral" (TSV = 0) was 5.56% and 20%, respectively. Based on the thermal sensation vote results of Site A and Site B, TSV = −2: Site A (42.59%) > Site B (26.67%); TSV = −1: Site A (50%) ≈ Site B (53.33%); TSV = 0: Site A (5.56%) < Site B (20%). Comparing the vote results of Site A and Site B, it can be seen that Site B is better than Site A in winter. Therefore, the thermal comfort ranking of four measuring sites in winter is Site C > Site B > Site A >Site D.

In summer, the results of thermal sensation vote (see Figure 8b) showed that the highest percentage of respondents (60%) felt "neutral" (TSV = 0) at Site D, so Site D had the best thermal comfort. The second highest percentage of respondents (52.73%) felt "neutral" at Site A (TSV = 0), so Site A had the second highest thermal comfort. Site B and Site C have a 20% vote for "neutral" (TSV = 0) and 34.55% and 41.82% for "warm" (TSV = 2), respectively. The result indicating Site B of thermal comfort is better than Site C. Combined with the thermal sensation vote results in summer, the thermal comfort ranking is Site D > Site A > Site B > Site C. The vote results of thermal sensation in winter and summer show that different types of landscape spaces have different effects on human thermal comfort, and the same landscape space has different thermal comfort in different seasons.

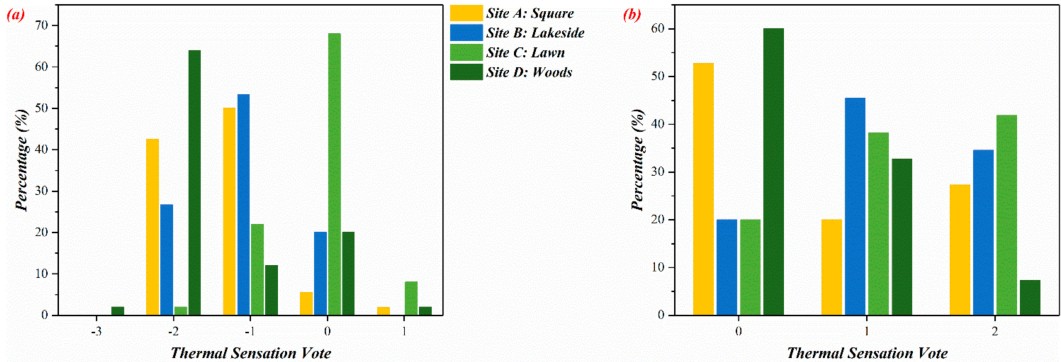

**Figure 8.** Thermal sensation vote (TSV) in four different measuring points. (a) Winter, (b) summer.

### 3.3.2. Meteorological Factors Preference Vote

In the winter, in the preference vote for air temperature (see Figure 9a), the percentage of respondents vote for "Higher" is Site D (98%) > Site A (94.44%) > Site B (64.44%) > Site C (46%).The ranking results are just the opposite of the thermal comfort vote results in winter (see Figure 8a), indicating that the relationship between air temperature and thermal comfort is extremely close. Similarly, the ranking of preferred air vote results in summer is also contrary to the thermal comfort vote results in summer (see Figure 8b), further illustrating that changes in air temperature have a significant impact on human thermal comfort. In the preference vote of relative humidity, except for Site A (drier: 70.39%) in winter, the other three measuring sites have the same preference for relative humidity regardless of winter and summer, and all hope to be unchanged (see Figure 9b). This result indicates that people are not sensitive to changes in humidity, which is consistent with the findings of Lai et al. [11]. However, in winter, the preference vote for relative humidity of Site A is different because most of the respondents in Site A are participating in square dancing, with a large amount of activity and sweat excretion. There is a need for sweat transpiration, so they hope that the relative humidity of the environment will be lower. An analysis of the respondents' preference for wind speed preference shows that in the winter, except for Site A (weaker: 83.33%), the majority of the other three surveyed respondents voted unchanged, the highest proportion of which was Site C (unchanged: 96%) (see Figure 9c). Site A has a relatively large wind speed in winter because the leaves of surrounding tree species fall off and the whole site is relatively open (SVF: 0.368), which affects the thermal comfort of the respondents, so the proportion of voters vote weaker is larger. In summer, the four measuring sites all hoped that the wind speed can be faster. The humid and hot environment in Chengdu makes the evaporation of human sweat slower, and the heat exchange process with the environment slower, so the thermal comfort is poor. The increase of wind speed can effectively increase the evaporation rate of human sweat, accelerate the process of heat exchange between human body and environment, and thus improve the thermal comfort of human body. The vote results of solar radiation preferences indicate that the four points in the winter are expected to be stronger for solar radiation. In the summer, except for Site C (weaker: 78%), the willingness to change solar radiation is stronger, the vote results of the other three points indicate that there is no strong willingness to change the solar radiation (see Figure 9d). The reason is that Site C (SVF: 0.389) is the most open place—lacking shelter and being exposed to direct sunlight for a long time—so the thermal comfort is poor, and the willingness to change is strong. The comprehensive respondents voted on the preference of meteorological parameters. The results show that air temperature is the most important factor affecting thermal comfort. Relative humidity has little effect on thermal comfort. Respondents hope to increase wind speed in summer and improve solar radiation in winter.

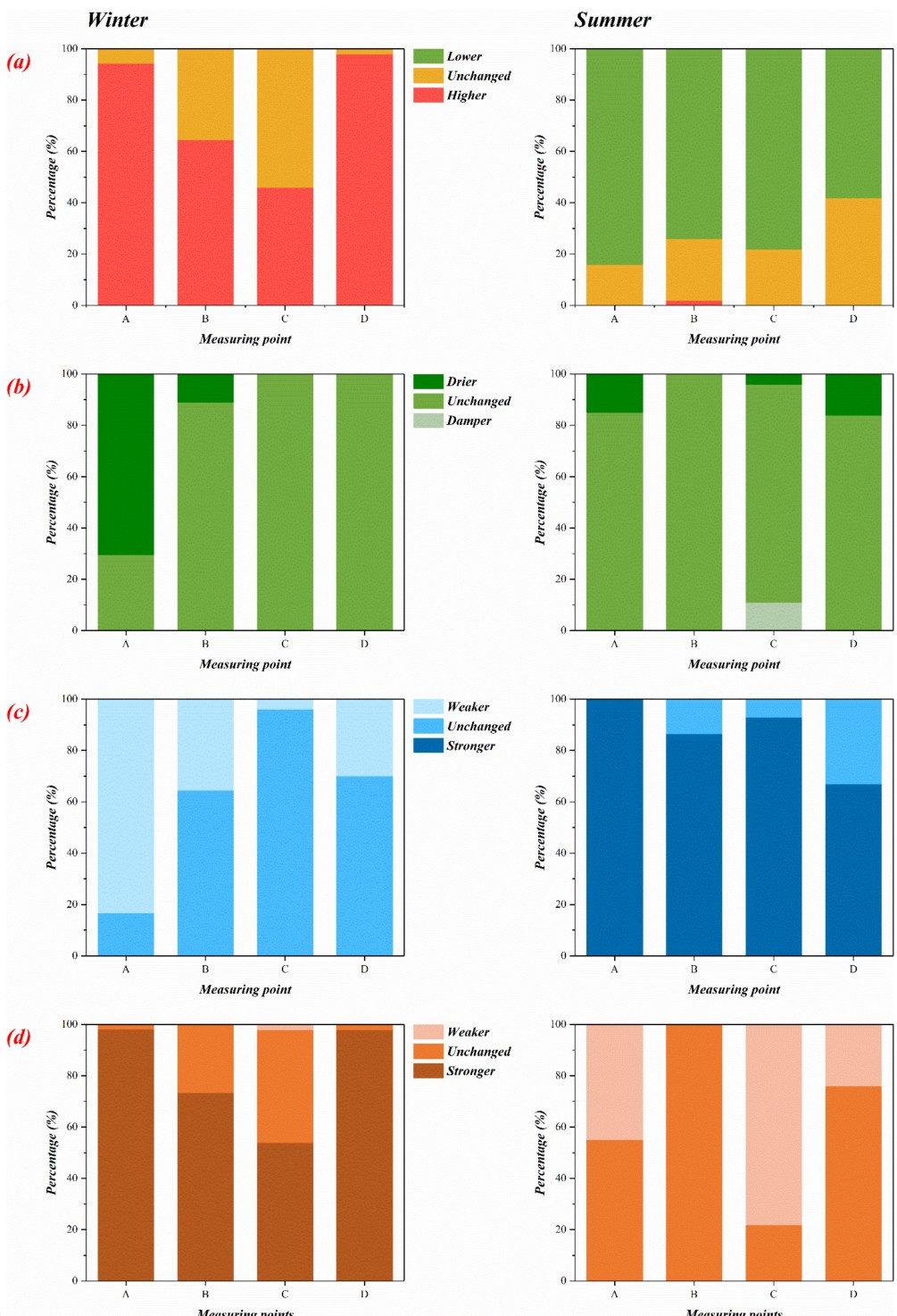

**Figure 9.** Preference votes for meteorological factors in the four different measuring points: (**a**) $T_a$ (air temperature), (**b**) *RH* (Relative Humidity), (**c**) *v* (wind speed), (**d**) *G* (global radiation).

### 3.3.3. Overall Thermal Comfort Vote (OTC)

Figure 10 shows the percentage of respondents who voted for overall thermal comfort (OTC) in each of the sites in winter and summer. The purpose of the overall thermal comfort vote is to fully validate the reasonableness of the thermal sensation vote. In winter (see Figure 10a), a very high percentage of respondents in Site B and Site C considered the thermal environment to be "neutral" (Site B: 95.56%, Site C: 100%). For the "uncomfortable" vote, Site A and Site D is 42.59% and 70%,

respectively. The overall thermal comfort of winter is ranked as Site C > Site B > Site A > Site D, and the ranking results are consistent with the results of the thermal sensation vote (see Figure 8a). The overall thermal comfort vote results for summer (see Figure 10b) are also consistent with the summer heat sensitivity vote results (see Figure 8b). The results show that there is a strong positive correlation between overall thermal comfort and thermal sensation.

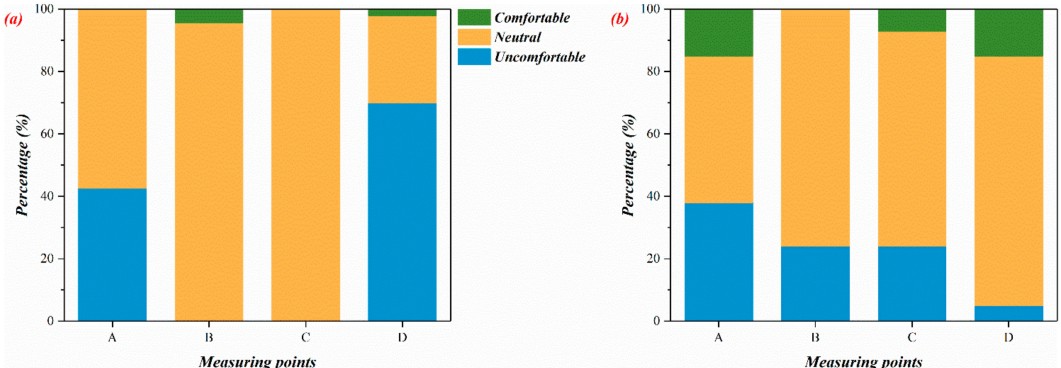

**Figure 10.** Overall thermal comfort votes (OTC) in four different measuring points. (**a**) Winter, (**b**) summer.

### 3.3.4. Thermal Acceptability Vote (TAV)

In order to further explore the impact of subjective psychological factors of urban residents on human thermal comfort. In the questionnaire survey, the respondents were asked questions about the thermal acceptability and related analysis. Figure 11 shows the percentage of respondents who voted for heat acceptability in each of the sites in winter and summer. In winter (see Figure 11a), the percentage of "acceptable" votes at each site was Site A: 55.5%, Site B: 100%, Site C: 98%, Site D: 34%. In winter, the thermal acceptability is ranked as Site C > Site B > Site A > Site D, and this ranking result is consistent with the results of the thermal sensation vote (see Figure 8a) and the overall thermal comfort vote (see Figure 10a). In summer, the results of thermal acceptability vote (see Figure 11b) were consistent with those of thermal sensation vote (see Figure 8b) and overall thermal comfort vote (see Figure 10b).Based on the analysis of the overall thermal comfort vote and the thermal acceptability vote, it can be seen that the results of the thermal sensation vote of the study are reasonable, and the ASHRAE seven-point scale determines that the thermal sensation is applicable to Chengdu.

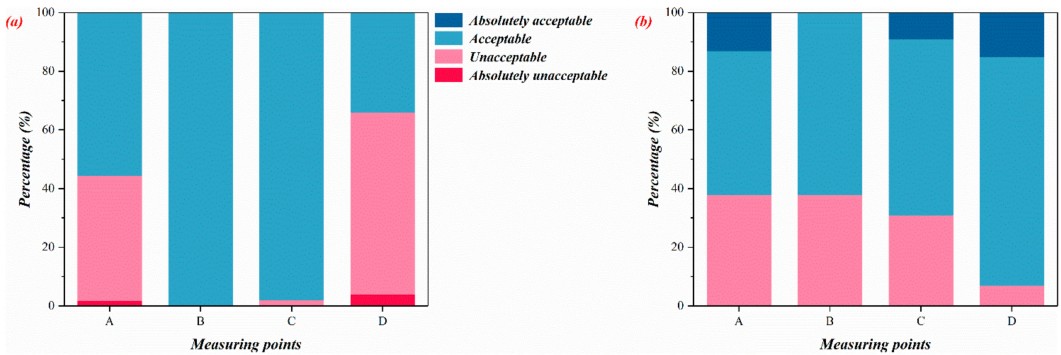

**Figure 11.** Thermal acceptability votes (TAVs) in four different measuring points. (**a**) Winter, (**b**) summer.

### 3.4. Thermal Indices

### 3.4.1. Physiological Equivalent Temperature (PET)

PET was determined by the clothing insulation, metabolic rate of interviewees and the corresponding meteorological parameters in a specific space in the RayMan model. The relationship

between the mean TSV and 1 °C PET was calculated, and linear fitting of PET and mean TSV in winter and summer respectively (see Figure 12). Obtain the following linear relationship:

$$\text{Winter: MTSV} = 0.08814 \text{ PET} - 1.10665 \quad (R^2 = 0.35646) \tag{2}$$

$$\text{Summer: MTSV} = 0.10653 \text{ PET} - 1.68792 \quad (R^2 = 0.81663) \tag{3}$$

When the MTSV is zero, the human body neutral temperature corresponding to PET is determined, so the neutral temperature PET is 12.6 °C in winter in Chengdu, and the neutral temperature of PET is 15.9 °C in summer. When the MTSV is between −0.5 and 0.5 (including −0.5 and 0.5), it is determined to be in the neutral PET range, so neutral PET range is 6.7–18.2 °C in winter in Chengdu, and neutral PET range is 11.2–20.5 °C in summer. In winter, the neutral PET temperature range in Chengdu is narrower than that of Tianjin (11–24 °C) [11] and Shanghai (15–29 °C) [41] but is better than Xi'an (13.3–23.6 °C) [33]. The range is wider and much wider than the range of Taiwan (26–30 °C) [46]. For neutral temperature PET range, its thermal stress spans four levels from "strong cold stress" to "no thermal stress" (see Table 3). In the summer, neutral PET temperature range of Chengdu also spans four levels in its size.

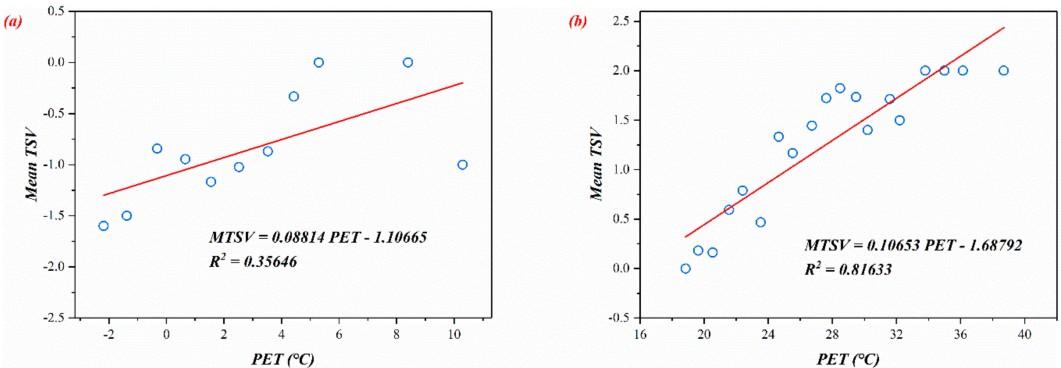

**Figure 12.** Correlation between PET and mean TSV. (**a**)Winter, (**b**) summer.

3.4.2. Universal Thermal Climate Index (UTCI)

UTCI is calculated in the same way as PET. The average meteorological parameters are collected every 15 minutes, and the clothing insulation and metabolic rate of respondents are input into the RayMan model at the corresponding time to calculate the corresponding UTCI value. Calculate the relationship between the mean TSV and 1 °C UTCI, and linearly fit the UTCI and the mean TSV in winter and summer (see Figure 13). Get the following linear relationship:

$$\text{Winter: MTSV} = 0.10206 \text{ UTCI} - 1.90917 \quad (R^2 = 0.56704) \tag{4}$$

$$\text{Summer: MTSV} = 0.18617 \text{ PET} - 4.74435 \quad (R^2 = 0.8203) \tag{5}$$

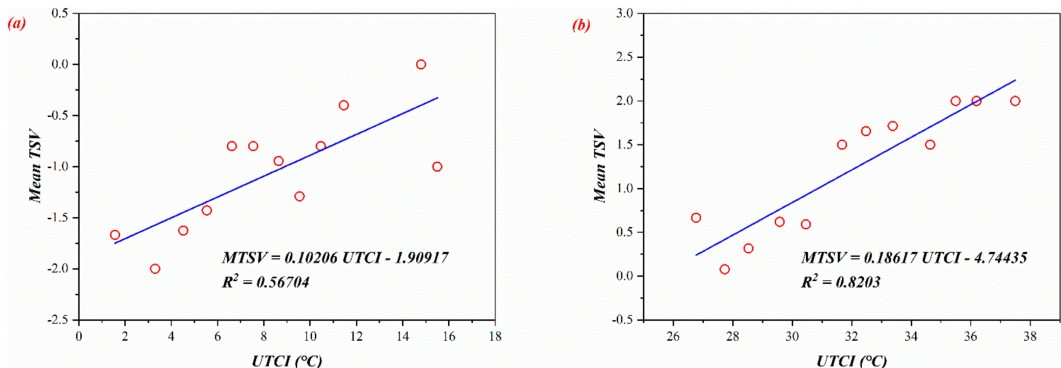

**Figure 13.** Correlation between UTCI and Mean thermal sensation vote (TSV). (**a**)Winter, (**b**) Summer.

Neutral temperature UTCI of Chengdu is (Winter: 18.7 °C; Summer: 25.5 °C).The neutral temperature range of UTCI in winter and summer in Chengdu is (Winter: 13.8 − 23.5 °C; Summer: 22.8 − 28.2 °C).In winter, the neutral temperature UTCI range in Chengdu belongs to "No thermal stress" in the thermal stress level. In summer, the neutral temperature UTCI range spans "No thermal stress" to "Moderate heat stress" and the temperature range Most of them still belong to "No thermal stress" (see Table 4). The neutral temperature range of UTCI is more reasonable than the PET neutral temperature range. Whether in winter or summer, a higher correlation was detected between UTCI and mean TSV by comparing the relationship between PET, UTCI and mean TSV (Winter: 0.56704 > 0.35646, Summer: 0.81663 > 0.8203). This indicates that UTCI can more accurately reflect the actual thermal sensation of urban residents in Chengdu than PET.

## 4. Discussion

### 4.1. Most Comfortable Type of Landscape Space

In winter, the thermal comfort ranking of the four types of landscape spaces is Lawn (Site C) > Lakeside (Site B) > Square (Site A) > Woods (Site D). However, the order is Woods (Site D) > Square (Site A) > Lakeside (Site B) > Lawn (Site C) in summer (See 3.3.1). The difference in the results of the two sorts indicates that different types of landscape space have different degrees of impact on human thermal comfort, which is consistent with the findings of Xu et al. [33] and Wang et al. [59]. It also shows that the thermal comfort of the same landscape space is different in different seasons, and similar research conducted by Li et al. [28], Fang et al. [37], and Shang et al. [38] also confirmed our research result.

Such research results show that in the actual park design, a single landscape space cannot be arranged in a concentrated manner, and the proportion of a certain landscape space should not be too large. Various landscape spaces should be reasonably configured so that the locations that meet the thermal comfort requirements of urban residents in different seasons.

### 4.2. Factors Affecting Human Thermal Comfort

By comparing the meteorological parameter data of Table 3 and the questionnaire data of Table 5 with the results of human thermal comfort in Section 3.3. It is found that the ranking of thermal comfort voting in winter is directly proportional to *Max $T_a$*, and other meteorological data (such as *Mean $T_a$*, *Mean RH*, etc.) have no significant linear relationship with the ranking results of thermal comfort voting, which indicates that the higher air temperature in winter has a more significant impact on thermal comfort. The ranking result of summer thermal comfort voting is inversely related to global solar radiation (*G*). The larger *G* is, the lower the order of thermal comfort voting results. It shows that in summer, solar radiation is the main factor affecting thermal comfort.

Moreover, after analyzing and comparing the results, it was found that the relative humidity (*RH*) and wind speed (*v*) had no significant effect on the results of the thermal sensory voting. The direct

result of the increase in solar radiation is the rise in air temperature. Therefore, based on the research results of winter and summer, it is found that air temperature is the most important factor affecting the outdoor thermal comfort of human body. In Section 3.3.2, the results of the study indicate that in terms of wind speed preferences, the interviewees at the four measurement points in summer hoped that the wind speed would be faster. Regarding the preference for relative humidity, whether in winter or summer, the majority of interviewees who voted for unchanged (winter 78.4% and summer 88.6%). It shows that the influence of wind speed on human thermal comfort is more important than relative humidity.

To sum up, air temperature ($T_a$) is the most important factor affecting human outdoor thermal comfort, and relative humidity ($RH$) is the least important. The results of this study are consistent with those of Lai et al. [11], Li et al. [31], Yao et al. [53], and Cheng et al. [62]. In the winter and summer research results, the main factors affecting thermal comfort were found to be inconsistent, indicating that changes in human thermal comfort are not controlled by a single factor, and more research is needed to understand the impact of different factors on human thermal comfort.

*4.3. Thermal Indices*

By comparing the research results of two thermal indicators of PET and UTCI, the results show that compared with UTCI, PET is more sensitive to changes in thermal pressure. Regardless of winter and summer, the temperature range of neutral PET, in thermal stress classification, spans four levels. UTCI only spans two levels, and most of its scope belongs to "no thermal stress". This shows that there is a significant difference in the width of the thermal stress distribution between the two indices. The width of the "no thermal stress" of PET is 5 °C (18 °C to 23 °C) and the width of the "no thermal stress" of UTCI is 17 °C (9 °C to 26 °C), the width of UTCI is much larger than that of PET. In this study, by comparing the relationship between PET, UTCI, and average thermal sensation vote (TSV), the significance of linear correlation was used to determine that UTCI is more suitable for outdoor thermal comfort evaluation in Chengdu than PET. A large number of studies have used this method to determine the applicability of the relevant thermal index in the study area [10–15,29,33,35,38,41,59,62]. However, the calculation methods of PET and UTCI are inconsistent, and the requirements for input data are also different. Therefore, further research is needed to better understand the essential difference between PET and UTCI.

*4.4. Study Limitations*

This study has achieved satisfactory results, but there are still some limitations. First, most of the interviewees in the questionnaire are over 45 years old, and the age of the respondents should be more evenly distributed. Second, our measurement period is only three days in the coldest and hottest months, less than 10% in winter and summer. Longer measurement cycles are needed, preferably throughout the season, to collect more comprehensive and persuasive data sets. Third, there are too few measuring sites for the landscape space to give a comprehensive and detailed overview of the different landscape spaces of the whole park. At the same time, the vegetation types in landscape space are not considered, and the effects of different plant species on the thermal environment are not consistent. Fourth, this paper focuses on the lack of garden structure (Chinese Pavilion, Chinese-style corridor, etc.) in different natural landscape spaces. Through field research, it is found that a large number of citizens will choose to rest and entertain in the interior of artificial landscape architecture.

## 5. Conclusions

In this study, the outdoor thermal comfort of four different landscape spaces in urban parks in China's hot summer and cold winter regions was investigated. Through the field monitoring of various meteorological parameters and questionnaires, and the data collected and analyzed, the outdoor thermal environment and human thermal comfort data of different landscape spaces in urban parks were obtained. Through comparative analysis, the following conclusions are obtained:

1.  Different types of landscape spaces have different effects on human thermal comfort, and the same landscape space has different thermal comforts in different seasons. In summer, the thermal comfort ranking for the four sites is Site D (Woods) > Site A (Square) > Site B (Lakeside) > Site C (Lawn); in winter, the thermal comfort ranking for the four sites is Site C (Lawn) > Site B (Lakeside) > Site A (Square)> Site D (Woods).

2.  Among various meteorological parameters, air temperature is the most important factor affecting outdoor thermal comfort, and the relative humidity has the least impact on thermal comfort. In urban parks, increasing wind speed during the summer and strengthening solar radiation during the winter can effectively improve outdoor thermal comfort. And urban residents in Chengdu are not sensitive to changes in the relative humidity of urban parks, and there is no strong willingness to change relative humidity.

3.  The overall thermal comfort vote (OTC) and the thermal acceptability vote (TAV) are closely related to the thermal sensation vote (TSV), and the results of the three questions are very similar. The ASHRAE seven-sites scale determines that the thermal sensation is applicable to the Chengdu area.

4.  The neutral temperatures of PET and UTCI are 12.6 °C and 18.7 °C in winter in Chengdu, respectively, and the neutral temperatures of PET and UTCI are 15.9 °C and 25.5 °C in summer, respectively. The neutral temperature ranges for PET and UTCI are 6.7 °C–18.2 °C and 13.8 °C–23.5 °C in winter, respectively, and the neutral temperature ranges for PET and UTCI are 11.2 °C–20.5 °C and 22.8 °C–28.2 °C in summer, respectively.

5.  Compare PET and UTCI to determine the accuracy of predicting outdoor thermal comfort for urban residents. The results show that UTCI is a better indicator for evaluating outdoor human thermal comfort in Chengdu.

The results of this study are of great significance to the landscape design of urban parks, which can help urban planners and landscape designers to improve the outdoor thermal environment and thermal comfort of cities on a scientific basis in future urban construction. In the future outdoor thermal comfort research, we should make up for the shortcomings of this study and sites out that we should understand the changes of human thermal sensation more comprehensively and accurately in order to provide more comfortable and healthy urban green space for urban residents.

**Author Contributions:** All the authors have read and approved the final manuscript. Data curation, D.W.; Formal analysis, D.W.; Funding acquisition, L.Z.; Investigation, D.W., Y.H., J.D. and Z.L.; Methodology, D.W.; Project administration, L.Z.; Resources, L.Z.; Software, D.W.; Supervision, L.Z., G.Z. and L.S.; Validation, D.W.; Visualization, D.W.; Writing – original draft, D.W.; Writing—review & editing, L.Z., G.Z. and L.S. All authors have read and agreed to the published version of the manuscript.

**Funding:** This research was partially funded by the key scientific research project of the Department of Education, Sichuan Province (NO.18ZA0370). This project is funded by the National Key R&D Program of China (2016YFC0700400), and the National Natural Science Foundation of China (No.51478280).

**Acknowledgments:** We would like to thank Yuyao Hou, Junfei Du, Jingyue Cheng, Zu'an Liu, Dong Wei, Haoru Liu, Qiong Shen and Chaoping Hou for their help in undertaking the field measurement and collect the data. Lili Zhang would like to thank the China Scholarship Council for supporting her stay as a Visiting scholar at RMIT University (201806915010).

**Conflicts of Interest:** The authors declare no conflict of interest.

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
