# Peer review of "Outdoor Thermal Comfort of Urban Park—A Case Study"

_sustainability, doi:10.3390/su12051961_

Round 1

Reviewer 1 Report

The manuscript is a noticeable scholar research investigating the outdoor thermal comfort of an urban park. Thus the subject of the manuscript refers to a significant issue, being studied extensively worldwide.

The manuscript meets the submission criteria of the journal and complies with the journal’s guidance. The abstract accurately summarizes the essential information of the paper. The experimental investigation is comprehensive and leads to reasonable outcomes.

Some weakness can be identified in the manuscript, which could be further improved. These are as follows:

In the abstract there have been used a lot of parenthesis for explanation, which can be avoided. The abstract should always be concise. The introduction could be improved by providing more references from the international literature, regardless the fact that the research was conducted in China. Lines 50,76,140,148,173, 202 etc.: Most of the manuscripts’ references are missing, there is an error. Methodology, section 2.3:  I would have wished to see more information about the methodical approach for conducting the questionnaire survey and about the definition and composition of the sample (i.e. the interviewees). It’s seems that a random sample of people was considered. The software program (Rayman) used needs to be added to the references, not only mentioned in the text. The presentation of the research outcomes should be further improved; for instance the results of the questionnaires can be presented through diagrams. Please check the text for English language and use more accurate terminology.For example: Line 483 : “…artificial landscape architecture (pavilions, corridors, etc.)” ...better terminology should be used. Line 42: instead of “..in weakening the urban heat island effect”, the international literature uses the term “mitigating the urban heat island effect”. Line 448: instead of “Filed monitoring ….”Field monitoring” is the more adequate. The term “urban” is excessively used. See for instance Lines 40-42 . “Urban park is not only a main place for outdoor activities of urban residents (citizens), but also an important part of urban green space system, which plays an important 41 role in weakening the urban heat island effect and improving the urban thermal environment (the outdoor thermal comfort).” Some suggestions are made in the parenthesis.

Reviewer 2 Report

In line 50, “see Error! Reference source not found.” should be changed. In line 51-54, the comparison of indoor and outdoor thermal comfort research is not correct. In line 58, full names of outdoor meteorological parameters (Ta, RH, Tg, v, G) should be given. In line 76, “see Error! Reference source not found.” should be changed. In line 98, the following reference can be added. Bin Yang, Thomas Olofsson, Gireesh Nair, Alan Kabanshi. 2017. Outdoor thermal comfort and human behavior pattern under subarctic climate of north Sweden - a pilot study in Umeå. Sustainable Cities and Society, 28, 387-397. How can you know thermal history of surveyed subjects? Perhaps they just left air-conditioned indoor environments, which may influence questionnaire survey results.

Reviewer 3 Report

There is a big problem with this paper for me as a reviewer - it is a complete mess as regards organisation, the frequent error messages which make it impossible to follow (presumably these often mean that references to figures are lost), massively long paragraphs of dense text poorly formatted tables with broken words, incorrect terms - like using "interviewer" instead of "interviewee and so on. This makes it difficult to see the merits of the paper and to follow the arguments and much of the results. In detail

Figure 1 is unnecessary. The illustration of the park needs to be bigger as at the size shown it is impossible to determine the layout. The relevance of the fish eye photos needs to be  explained in more detail. You also need to explain how Chendu, in a sub-tropical climate with warm winters according to Koppen is in a cold winter zone - this makes no sense!

At this point therefore, I request that the authors reformat the paper, make sure that the English is acceptable, make sure that the figures are referred to in the text and are clear, improve the formatting of the tables and break up the really long  paragraphs. Then I will be able to give a proper review

Round 2

Reviewer 2 Report

The manuscript can be accepted.

Author Response

Thank you very much for your comments, which were extremely valuable in helping to improve the paper quality. According to your comments, we tried our best to make the appropriate revision, and make this paper can be acceptable. We appreciate your hard work and thanks for your help.

Reviewer 3 Report

This is much improved now, with all the issues resolved which made the paper difficult to read. It is very dense with a lot of data and many results which make it difficult to be able to synthesise it all as one reads it. The results are presented logically but the paper completely lacks a discussion which should bring the results together and relate them to the literature and previous research findings - calling the main section results and discussion, when you have such a lot of results is not ideal as the discussion, such as it is, is lost among the massive detail. The results suddenly stop and then we have the conclusions. There is no real summary and no discussions of implications for planners or designers. The paper really just presents the results without a lot of comment. It should refer back formally to the objectives as stated so that the aims of the research can be seen to have  been properly met.

Next time, after adding the discussion, can you present a properly formatted paper with no track changes as these also made it difficult to follow inplaces and as the file is a pdf it is not possible to switch them of myself.

The English needs some attention - you miss the definite or indefinite article in many sentences. Get it checked properly.

Round 3

Reviewer 3 Report

I am afraid I stand by my comments - the very fact that you have so many results means that a separate discussion section is essential in order to connect them together and to make sense of the big picture and not just each specific element. Explaining the results as you go is necessary of course and this is what you do but at the end the reader needs to see how the whole picture comes together and the linkages between the different aspects presented separately in the results need to be pointed out. You conducted the research and you know in great detail all the different aspects and results but for a reader not so familiar and new to the work the questions I raise are also likely to be raised in their minds. This is one of the roles of a reviewer, coming to the work as an outsider. Therefore I must request that you provide additional discussion as abridge between the results section and the conclusions which considers how the results fit together - the conclusions still maintains a separation of the different results section.
